# Effects of Whole-Plant Corn Silage on Growth Performance, Serum Biochemical Indices, and Fecal Microorganisms in Hezuo Pigs

**DOI:** 10.3390/ani14050662

**Published:** 2024-02-20

**Authors:** Xitong Yin, Pengfei Wang, Zunqiang Yan, Qiaoli Yang, Xiaoyu Huang, Shuangbao Gun

**Affiliations:** 1College of Animal Science and Technology, Gansu Agricultural University, Lanzhou 730070, China; yinxitong0324@163.com (X.Y.); wangpf815@163.com (P.W.); yanzq@gsau.edu.cn (Z.Y.); yangql0112@163.com (Q.Y.); huanghxy100@163.com (X.H.); 2Gansu Diebu Juema Pig Science and Technology Backyard, Gannan 747400, China; 3Gansu Research Center for Swine Production Engineering and Technology, Lanzhou 730070, China

**Keywords:** Hezuo pigs, whole-plant corn silage, growth performance, serum biochemical indices, fecal microorganisms

## Abstract

**Simple Summary:**

In recent years, with the expansion of pig husbandry, the competition for grain between its use as feedstuffs for animals and as foodstuffs for humans has become increasingly prominent. Traditional feeds such as corn, soybean, and wheat bran are no longer sufficient to meet the growing needs of the pig industry. Consequently, it is imperative to develop novel, unconventional feedstuffs. Whole-plant corn silage has the advantages of high energy, easy digestion, good palatability, aromatic taste, long preservation time, and low cost. In this study, we found that adding the proper amount of whole-plant corn silage to the diet can not only promote the growth of Hezuo pigs, but also replace part of the traditional feed. Whole-plant corn silage has a broad application prospect in the Hezuo pig diet.

**Abstract:**

In this study, we investigated the effects of the dietary inclusion of different proportions of whole-plant corn silage on growth performance, serum biochemical indexes, and intestinal microorganisms in Hezuo pigs. Thirty-two two-month-old Hezuo pigs (body weight: 7.88 ± 0.81 kg) were randomly divided into four groups of eight pigs (half male, half female) each. The control (CON) group received a basal diet, while the three experimental groups were fed the basal diet, part of which had been replaced with 5%, 10%, and 15% whole-plant corn silage, respectively. The experiment lasted for 127 days, including 7 days of pre-testing and 120 days of formal testing. At the end of the experiment, blood and fecal samples were collected. Compared with the CON group, the feed-to-gain ratio was significantly lower in the 10% test group (*p <* 0.05), whereas the total protein, albumin, triglyceride, and glucose contents were significantly higher (*p <* 0.05). No significant differences in total cholesterol, high-density lipoprotein cholesterol, low-density lipoprotein cholesterol, creatinine, urea, aspartate aminotransferase, and alanine aminotransferase were observed among the groups (*p >* 0.05). The addition of whole-plant corn silage to the diet significantly increased alpha diversity in the pig gut based on 16S rRNA gene sequencing. The principal coordinate analysis results showed significant clustering of the different groups (*p <* 0.05). At the phylum level, the addition of whole-plant corn silage to the diet significantly decreased (*p <* 0.05) the relative abundance of Firmicutes and significantly increased (*p <* 0.05) that of Bacteroidetes. At the genus level, the relative abundance of *Streptococcus* significantly decreased (*p <* 0.05) with increasing silage supplementation levels, whereas species diversity significantly increased (*p <* 0.05). In conclusion, 10% is the recommended inclusion ratio for whole-plant corn silage in the diets of pigs.

## 1. Introduction

Pig farming is the key basic industry in traditional Chinese agriculture [1]. The Hezuo pig breed, distributed in the highlands of the northern Tibetan plateau, is unique to Gansu, China. It has good roughage tolerance, good meat quality, excellent resistance to harsh conditions including severe cold, tolerance to strong sunlight, and is well-adapted to living in areas with an average altitude of 3000 m [2]. It is an excellent small lean pig with thin skin and delicious meat. Hezuo pork has high nutritional value, wide and uniform marbling distribution, and a unique flavor. Accordingly, Hezuo pork has notable marketing potential, and improving the productivity of the Hezuo pig industry would be of great benefit to local Tibetan people [3].

China is currently facing a serious food shortage problem. In 2021, the total grain supply in China was 870 million tons, 450 million of which were consumed by the farming industry, accounting for 51.72% of the total consumption [4]. Additionally, competition between humans and animals for food has greatly affected China’s food security [5]. Given these challenges, effective measures, such as the development of animal feed resources, are needed to stabilize the development of China’s farming industry and solve the issue of feed resource shortage. Silage refers to the cutting of fresh green feed, its sealing in containers, and subsequent fermentation by microorganisms, yielding feed with specific aromas and high nutritional value. Over recent years, whole-plant corn silage has become a common feed for ruminants such as dairy cattle, beef cattle, and sheep worldwide because of its good palatability, long-term preservability, and resistance to weather [6]. In 2020, the maize silage harvesting area in China was 21,987 hm^2^, which was a significant increase of 32.04% compared with the 16,652 hm^2^ recorded in 2016. The maize silage planting area in China is predicted to reach 67,000 hm^2^ in 2030 [7]. Whole-plant corn silage has great potential for development in China. Wang et al. [8] found that adding an appropriate amount of whole-plant silage corn to the diet improved the growth performance of young geese aged 28 to 70 days, but reduced the slaughter performance. Ren et al. [9] reported that the dietary addition of more than 20% fresh whole-plant corn silage decreased the average daily gain (ADG) of Bamei pigs. Wang et al. [10] replaced part of the diet of growing finishing pigs (60–110 kg) with whole-plant corn silage, and found that pork tenderness was improved while also meeting the nutritional requirements of the animals. These observations indicated that whole-plant corn silage can be used as feed material for pigs.

The effects of whole-plant corn silage have mainly been assessed in ruminants such as dairy cows and beef cattle, and little is known about how they affect monogastric animals. Hezuo pigs are characterized by high roughage tolerance, forage and vegetation intake, and digestive and absorptive capacity [3]. Whole-plant corn silage is generally added wet to ruminant feeds, which requires the use of labor and machinery. Additionally, in winter, due to cold weather, fresh whole-plant corn silage tends to clump and is not easily accessible, which increases farming costs. In this study, whole-plant corn silage was first sun-dried and crushed, and only then was it added to the diets of Hezuo pigs. Subsequently, the effects of different addition levels (5%, 10%, and 15%) of the silage on growth and development, serum biochemical indices, and fecal microorganisms of Hezuo pigs were investigated. The aim of the study was to provide a theoretical basis for the utilization of whole-plant corn silage in Hezuo pig diets and identify the most suitable inclusion ratio.

## 2. Materials and Methods

### 2.1. Experimental Materials

Whole-plant corn silage was provided by a dairy plant in Baiyin Yongdong Farm (118°72′ E, 35°33′ N, Lanzhou, Gansu Province, China). The levels of its conventional nutrient contents are shown in Table 1.

### 2.2. Time and Location of the Experiment

The experiment was carried out from June to October 2022 at Gansu Xigu Bailv Tourism Farm (103°63′ E, 36°08’ N, Lanzhou, Gansu Province, China).

### 2.3. Experimental Design and Animals

In total, 40 healthy 2-month-old Hezuo pigs with similar body weight were randomly selected for a 7-day pre-experiment in which they were fed a basal diet. At the end of the pre-experiment, 32 pigs (body weight: 7.88 ± 0.81 kg) were randomly divided into 4 groups (a control [CON] group and 3 experimental groups) of 8 pigs each (half male, half female). Pigs in the CON group received a basal diet, while those in the experimental groups were fed the basal diet, part of which had been replaced with 5%, 10%, and 15% whole-plant corn silage, respectively. The whole experiment lasted 120 days. During this time, all the pigs were kept in identical single cages under the same feeding management conditions and were allowed ad libitum access to food and water.

### 2.4. Experimental Diet

Whole-plant corn silage was dried and crushed to a particle size of 1.65 mm and mixed according to the feed formula. The whole-plant corn meal silage replaced 5%, 10%, or 15% of the basal diet in the experimental groups. Pig feed is configured according to pig energy requirements [11]. The composition and nutrient contents of the diets are shown in Table 2.

### 2.5. Measurement Indicators and Methods

#### 2.5.1. Growth Performance Indexes

At the beginning and end of the experiment, the pigs were weighed after a 12 h fast (water was freely available), and the values were taken as the initial body weight (IBW) and final body weight (FBW), respectively. During the test period, the feed consumption of each pig was recorded, and the average daily feed intake (ADFI), ADG, and feed-to-gain (F/G) ratio were calculated.

Growth performance indicators were calculated as follows:

ADG = (FBW − IBW)/the number of days in the experiment.

ADFI = feed consumption/the number of days in the experiment.

F/G ratio = feed consumption/total weight gain.

#### 2.5.2. Measurement of Serum Biochemical Indicators

At the end of the feeding test, 10 mL of blood was collected from the anterior vena cava of six pigs (half male, half female) randomly selected from each group, left to stand for 15 min, and then centrifuged at 4000× *g* rpm for 15 min (Centrifuge 5810R, Eppendorf, Hamburg, Germany). The obtained serum (supernatant) was stored for testing. The serum biochemical indicators measured included total protein (TP), albumin (ALB), total cholesterol (TC), triglyceride (TG), high-density lipoprotein cholesterol (HDL-C), low-density lipoprotein cholesterol (LDL-C), creatinine (Cr), urea nitrogen (urea), glucose (GLU), aspartate aminotransferase (AST), and alanine aminotransferase (ALT).

#### 2.5.3. Sequencing of Fecal Microbiota

At the end of the experiment, fresh feces from each group of 6 pigs (half male, half female) that were not in contact with the ground were collected using sterile cotton swabs and were divided into 1-mL sterile freezing tubes, frozen in liquid nitrogen, returned to the laboratory, and stored at −80 °C.

The samples were sent to Shanghai Personal Biotechnology Co., Ltd. (Shanghai, China) on dry ice and assayed according to the company’s test procedure, as follows: DNA was extracted using an OMEGA Soil DNA Kit (M5635-02; Omega Bio-Tek, Norcross, GA, USA), and its purity and content were assessed using 0.8% agarose gel electrophoresis and a NanoDrop (NC2000, Thermo Fisher Scientific, Waltham, MA, USA).

The V3–V4 region of the bacterial 16S rRNA genes was PCR-amplified using the universal primer pair 338F (5′-ACTCCTACGGGAGGCAGCA-3′) and 806R (5′-GGACTACHVGGGTWTCTAAT-3′). The cycling conditions consisted of an initial denaturation step at 98 °C for 5 min, followed by 25 cycles of denaturation at 98 °C for 30 s, annealing at 53 °C for 30 s, and extension at 72 °C for 45 s, with a final extension of 5 min at 72 °C. PCR amplicons were purified with VAHTSTM DNA Clean Beads (Vazyme, Nanjing, China) and quantified using the Quant-iT PicoGreen dsDNA Assay Kit (Invitrogen, Carlsbad, CA, USA). After quantification, the amplicons were pooled in equal amounts, and paired-end (2 × 250 bp) sequenced using the Illumina NovaSeq platform with the NovaSeq 6000 SP Reagent Kit (500 cycles) at Shanghai Personal Biotechnology Co., Ltd. The sequencing data were deposited in the Sequence Read Archive (SRA) of NCBI under accession number PRJNA1056331.

### 2.6. Data Analysis

The 16S rRNA gene sequencing data were analyzed using Quantitative Insights Into Microbial Ecology (QIIME), version 1.8.0 (http://qiime.org/ (accessed on 21 August. 2023).), and the R package, version 3.5.1 (https://www.r-project.org/ (accessed on 21 August 2023)). To determine microbial community structural variation across samples, beta-diversity was analyzed using UniFrac distance metrics and visualized via principal coordinate analysis (PCoA) plots. Differences in UniFrac distances among the groups were determined by analysis of similarities (ANOSIM). Heatmap was plotted using R package 3.5.1.

Data relating to microbial alpha-diversity indices (Shannon, Simpson, Observed, and Chao1), growth performance, serum biochemical indices, and the abundance of fecal microorganisms were analyzed by one-way ANOVA followed by Duncan’s post hoc test in SPSS software (version 23.0; IBM Corp., Armonk, NY, USA). The data are expressed as means ± standard deviation. A *p*-value < 0.05 was considered significant.

## 3. Results

### 3.1. The Effect of Whole-Plant Corn Silage on the Growth Performance of Hezuo Pigs

The differences in ADFI and ADG at the end of feeding among the four groups were not significant (*p >* 0.05). The F/G ratio in the 10% supplementation group was significantly lower than that of the CON group (*p <* 0.05); however, those of the 5% and 15% groups did not differ significantly from that of the CON group (*p >* 0.05) (Table 3).

### 3.2. The Effect of Whole-Plant Corn Silage on Serum Biochemical Indices of Hezuo Pigs

There were no significant differences in TC, HDL-C, LDL-C, Cr, urea, AST, or ALT contents among the groups (*p >* 0.05). The TP, ALB, TG, and GLU contents were significantly higher in the 10% whole-plant corn silage supplementation group than in the CON group (*p <* 0.05); in contrast, no significant differences in the levels of these indices were found between the 5% or 15% group and the CON group (*p >* 0.05) (Table 4).

### 3.3. The Effects of Whole-Plant Corn Silage on the Fecal Microbiota of Hezuo Pigs

#### 3.3.1. Alpha-Diversity

To investigate the effects of the addition of different proportions (0%, 5%, 10%, and 15%) of whole-plant corn silage on the fecal microflora of Hezuo pigs, we performed 16S rRNA gene sequencing of fecal samples of all the groups. After quality control, an average of 74,828 sequence reads were obtained from each sample. Rarefaction curve analysis indicated that almost all microorganisms were detected in the feces of Hezuo pigs (Figure 1a). The total number of operational taxonomic units (OTUs) was 20,879, with 1046 being shared among all the groups. A total of 3885 species were unique to the CON group, while 3528, 4816, and 5105 species were unique to the 5%, 10%, and 15% addition groups, respectively (Figure 1b). The Shannon and Simpson index values increased with increasing whole-plant corn silage supplementation levels (Table 5).

#### 3.3.2. Comparison among Microbial Communities (Beta-Diversity)

Unweighted UniFrac distances were used to perform a PCoA and derive principal coordinates (Figure 2a). The results showed significant additive clustering with increasing corn silage supplementation levels, with Hezuo pig microbial communities appearing to become more similar at increasing supplementation doses. However, no differences in the dispersal of individual microbial communities were found across the different supplementation groups based on weighted UniFrac analysis (Figure 2b).

#### 3.3.3. Phylogenetic Composition of Fecal Microorganismal Communities

The dominant flora at the phylum level are shown in Figure 3a. Firmicutes was the most abundant phylum, followed by Bacteroidetes. Table 6 shows the taxonomic composition of the top 10 gut microbial communities at the phylum level. The abundance of Firmicutes decreased significantly with increasing addition levels of whole-plant corn silage (*p <* 0.05), whereas that of Bacteroidetes, Proteobacteria, and Tenericutes showed the opposite trend (*p <* 0.05). The abundance of Verrucomicrobia was significantly higher in the 15% supplementation group than in the CON or 10% addition groups (*p <* 0.05). The abundance of Cyanobacteria was significantly higher in the 5% supplementation group than in the CON, 10%, and 15% supplementation groups (*p <* 0.05). The dominant flora at the genus level are shown in Figure 3b. *Streptococcus* displayed the highest relative abundance, followed by *Lactobacillus*. The taxonomic composition of the top 10 gut microbial communities at the genus level is presented in Table 7. The relative abundance of *Streptococcus* decreased significantly with increasing addition levels of whole-plant corn silage (*p <* 0.05). Meanwhile, the relative abundance of *Ruminococcus* was significantly higher in the 15% corn silage addition group than in the CON or 5% supplementation groups (*p <* 0.05). The relative abundance of *Oscillospira* was significantly higher in the 5% supplementation group than in the three other groups (*p <* 0.05). The relative abundance of *Roseburia* was significantly lower with the addition of 5% corn silage to the diet than when 15% corn silage was added (*p <* 0.05). The relative abundance of *SMB53* was significantly lower in the CON group than in the 10% and 15% supplementation groups (*p <* 0.05). Figure 4 shows the heat-maps of the taxonomic composition of the fecal microbiota of Hezuo pigs at the phylum and genus levels among the different groups.

## 4. Discussion

### 4.1. The Effects of Whole-Plant Corn Silage on the Growth Performance of Hezuo Pigs

Over the last few decades, the use of whole-plant corn silage by livestock producers has become increasingly common, owing to its generally good ensiling characteristics, high biomass yield, relatively high metabolizable energy content, and varied range of growth periods [12]. Cristina et al. [13] found that corn silage had good palatability, and its addition to feed did not affect dry matter intake in pigs. Lyu et al. reported that the addition of appropriate proportions of corn silage to the feed of pigs increased the abundance of gastrointestinal microbial communities, promoted digestion and absorption, and significantly increased ADFI and ADG. The authors further reported that the addition of 10% whole-plant maize silage to the ration resulted in the highest final live weight (370.7 kg) among the supplementation levels tested [14]. Adding more than 40% sun-dried corn silage to a feed exerted a significantly negative effect on the final weight of the animals. At a low percentage, maize silage added to the feed increased bacterial species richness and promoted digestion and absorption, thereby decreasing the occurrence of gastrointestinal diseases [15,16]. In our study, we found that the addition of 5%, 10%, and 15% whole-plant corn silage did not affect the ADFI or ADG of Hezuo pigs; however, at the 10% addition level, the F/G ratio was significantly decreased compared with that of the control, similar to the findings of Lyu et al. [14]. Weng et al. [17] found that when the dietary crude fiber level in the diet of growing-finishing pigs increased from 3% to 6%, the F/G ratio decreased significantly. In line with these results, we observed that the F/G ratio decreased significantly when the crude fiber content of the feed increased from 3.45% to 5.85%. This demonstrated that the dietary inclusion of 10% whole-plant corn silage has a promotive effect on the growth performance of Hezuo pigs and represents the best addition level.

### 4.2. The Effects of Whole-Plant Corn Silage on Serum Biochemical Indices of Hezuo Pigs

When animals are fed different feeds, their metabolism changes, which affects biochemical indicators, such as blood metabolite and nutrient contents. Accordingly, biochemical indicators, to a certain extent, reflect the metabolic status of an animal. Additionally, changes in energy and protein (or amino acid) levels in the diet usually result in alterations in the contents of blood lipid and protein metabolism-related indicators [18]. The concentrations of GLU, TG, Cr, TP, ALB, and urea are the blood biochemical parameters most commonly used to assess the metabolic profile of carbohydrates, fats, and proteins; when these concentrations vary within normal ranges, higher values imply better energy and protein nutritional profiles [19]. Su et al. found that serum cholesterol and HDL-C contents significantly increased in Yuejiang donkeys with the addition of increasing proportions of whole-plant corn silage to the feed [20]. In our study, we found that the TP and ALB contents were significantly higher in pigs fed a diet supplemented with 10% whole-plant corn silage than in control animals; the TP and ALB contents were also higher in the other corn silage supplementation groups than in the CON group, but the difference did not reach significance. These results suggested that the addition of whole-plant corn silage to the ration enhances protein metabolism, which is similar to the results of Su et al. [20]. As the supplementation level increased, the crude fiber content also increased, while the energy level of the ration and the TC content decreased. However, the changes were not significant, indicating that the addition of whole-plant corn silage to the diet did not significantly affect the serum TC content of Hezuo pigs. The TG content was significantly higher in the 10% supplementation group than in the CON group, an effect that was associated with a higher dietary fiber content, which promotes the absorption of GLU and fatty acids through the small intestine. Wang et al. [8] found that the dietary inclusion of whole-plant corn silage decreased the TG content in serum of goslings, whereas, in our study, the TG content exhibited the opposite trend. This disparity may be related to differences in the digestive structure of the animals. Although the variation in HDL-C and LDL-C contents among the four groups was not significant, the serum HDL-C content of Hezuo pigs was higher in both the 5% and 10% whole-plant corn silage supplementation groups than in the CON group, indicating that the addition of whole-plant corn silage to the ration of Hezuo pigs tended to improve their lipid profile. This was in line with that reported by Brambillasca et al. [16] and may explain why the growth performance of pigs fed whole-plant corn silage was better than that of control animals. The main reason for the elevated serum GLU levels in Hezuo pigs may be related to the rate of fattening and backfat thickness and the fact that the glycerol and fatty acids needed for TG synthesis are mainly generated from GLU [21]. The GLU present in the blood of animals is obtained by two means. One involves GLU absorption through the intestinal wall, and the other is the fermentation of dietary fiber inside the digestive tract, which generates a large amount of volatile fatty acids that are consumed in the liver to generate GLU [22]. The addition of whole-plant corn silage resulted in higher nutrient utilization by the Hezuo pigs, with a corresponding increase in GLU content, which also contributed to the high TG content. This is one of the causes of high FBW. The detection of serum enzyme activities in animals can serve as an important indicator for the diagnosis of chronic diseases and animal tissue and organ dysfunction, as well as the health of the liver and the degree of stress response [23]. In this study, we found that the addition of whole-plant corn silage at different concentrations did not significantly affect serum enzyme activities in Hezuo pigs; nevertheless, the levels of these enzymes were all higher in the supplementation groups than in the CON group. AST and ALT enzyme activities did not differ significantly among the various groups and were within the normal range. This indicated that the addition of whole-plant corn silage did not adversely affect the livers and hearts of the pigs in our study.

In summary, the addition of 10% whole-plant corn silage improved protein metabolism and the lipid profile in Hezuo pigs. At the same time, this is also one of the main reasons for the lower F/G ratio and higher FBW than other groups.

### 4.3. The Effect of Whole-Plant Corn Silage on the Fecal Microbiota of Hezuo Pigs

The fecal microflora plays a key role in gut development. Studies have shown that the composition and abundance of gut microbes are affected by many factors, such as individual differences among animals [24], the composition of diets, and the ratio of fineness to roughness [25]. Under physiological conditions, the microflora in the animal intestinal tract is in a state of dynamic equilibrium with the host. The latter provides a habitat and nutrition for the flora, while the flora plays an important role in the host’s nutrient metabolism, nutrient sensing, immune function, and organismal health [26]. Animals are sterile before birth; after birth, through breathing, feeding, body contact, and other activities, they gradually establish a stable intestinal microbial system [27]. Wang et al. [28] found that the addition of different proportions of whole-plant corn silage to the diet did not affect the major bacterial phyla in the cecum of geese, but did affect the overall microbial composition, as well as the abundance of some bacterial phyla and genera; moreover, they reported that the abundance, uniformity, and diversity of gut microorganisms in the cecum of geese increased in a whole-plant corn silage concentration-dependent manner. Tang et al. [29] demonstrated that the dietary provision of fermented feeds improved the gut microbial composition, as evidenced by a significant decrease in the abundance of pathogenic bacteria (Proteobacteria phylum, *Escherichia–Shigella* genus) and a significant increase in that of beneficial bacteria (Firmicutes phylum, *Clostridium* genus). Zheng et al. [30] research found that microbial diversity is related to pig growth performance. Our results showed that the addition of whole-plant corn silage to the diet increased alpha diversity in Hezuo pigs. The Shannon index increased markedly with increasing inclusion ratios, while the Simpson index was significantly higher in the supplementation groups than in the CON group. This suggests that a more diverse microbial community and beneficial bacteria were present in the manure of the whole-plant corn silage group, which may have promoted the health status of pigs or improved the metabolic level, thereby helping to improve the feed conversion rate. These results are in line with those reported by Tang et al. and Chao et al. [29,31] in studies involving the addition of fermented feed to the rations of swine. The PCoA results showed that there were differences among the bacterial communities of the different whole-plant corn silage supplementation groups, suggesting that varying the whole-plant corn silage inclusion ratio can affect the fecal microbial composition of Hezuo pigs.

Numerous studies have shown that Firmicutes and Bacteroidetes are the dominant phyla in swine fecal microflora with relative abundances higher than 90%, and both phyla are important constituents of the microbiota. Members of both phyla can promote fiber decomposition and carbohydrate degradation, which are related to nutrient absorption and are beneficial to organismal health [32,33,34]. Differences in the composition of the intestinal microflora of pigs are also closely related to individual growth performance [35]. In our study, *Firmicutes* and *Bacteroidetes* were the dominant phyla in the fecal flora of the pigs, consistent with the results of the above-mentioned studies [36]. Moreover, the addition of whole-plant corn silage to the forage diet promoted the proliferation of Bacteroidetes, whereas the abundance of Firmicutes decreased with increasing corn silage addition levels. This indicated that the inclusion of whole-plant corn silage in the feed increased the digestive metabolism of the pigs, leading to weight loss. The addition of whole-plant corn silage to the ration influenced the abundances of *Proteobacteria*, *Tenericutes*, *Verrucomicrobia*, and *Cyanobacteria*. In summary, the addition of whole-plant corn silage to feed enhances immune function, improves intestinal microflora, changes the relative abundance of flora, and promotes the intestinal absorption of nutrients. Currently, infectious diseases caused by bacteria pose a significant challenge to the swine industry. For instance, *Streptococcus suis* infection causes septicemia, lymph node enlargement, arthritis, and other symptoms in pigs. In addition, *Streptococcus suis* is a common zoonotic bacterium that, in humans, can cause pneumonia, encephalitis, septicemia, endocarditis, fibrositis, and impaired prognosis of hearing [37,38]. In the present study, we found that the number of *Streptococcus* spp. decreased significantly with increasing whole-plant corn silage addition levels, indicating that the inclusion of whole-plant corn silage in the feed can reduce the abundance of some species of pathogenic bacteria in pigs, thereby reducing the incidence of disease in the animals and improving the efficiency of swine farming. The relative abundances of the genera *Ruminococcus, Oscillospira, Roseburia*, and *SMB53* were significantly increased with the addition of whole-plant corn silage to the diet, indicating that the inclusion of this silage tended to exert an ameliorative effect on the intestinal microorganisms of Hezuo pigs, which, in turn, promoted their growth.

## 5. Conclusions

Adding whole-plant corn silage to the diet did not exert adverse effects on Hezuo pigs. The addition of corn silage improved the intestinal microflora of the animals, characterized by a significant increase in bacterial diversity at the phylum and genus levels. The dietary inclusion of 10% whole-plant corn silage exerted a notable positive effect on growth performance, protein metabolism, and blood lipids in Hezuo pigs. Accordingly, we recommend 10% as the optimal dosage of whole-plant corn silage to be added to the rations of pigs.

## Figures and Tables

**Figure 1 animals-14-00662-f001:**
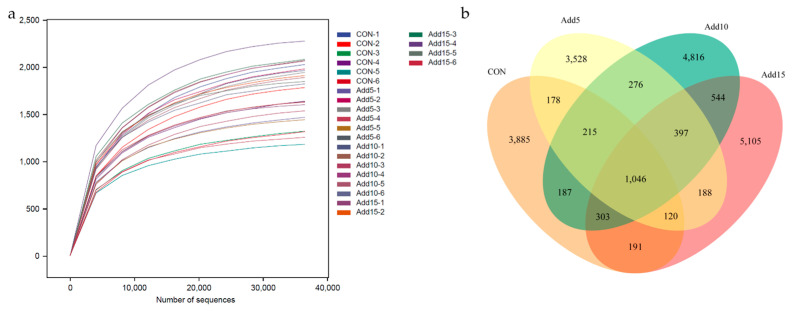
The effects of the addition of different proportions (0%, 5%, 10%, and 15%) of whole-plant corn silage on the fecal microflora of Hezuo pigs. (**a**) Microorganism rarefaction curves based on observed indices were used to assess the depth of coverage of each sample (samples are distinguished by different colored lines). (**b**) A Venn diagram showing the fecal microbial operational taxonomic unit (OTU) composition of the control (CON) group and the 5%, 10%, and 15% whole-plant corn silage supplementation groups. The samples were obtained from six Hezuo pigs in each group before the end of the test.

**Figure 2 animals-14-00662-f002:**
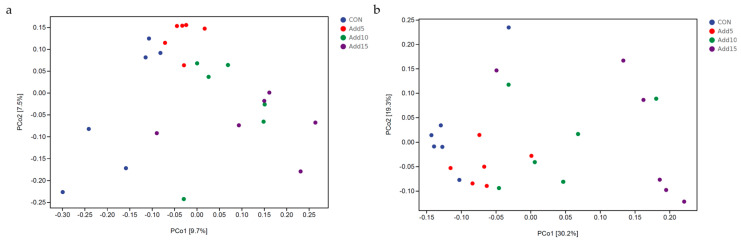
Development of gut microbial operational taxonomic units (OTUs) with increasing whole-plant corn silage supplementation levels. (**a**) Unweighted UniFrac distance based on the relative abundance of microbial OTUs. (**b**) Weighted UniFrac distance based on the relative abundance of microbial OTUs.

**Figure 3 animals-14-00662-f003:**
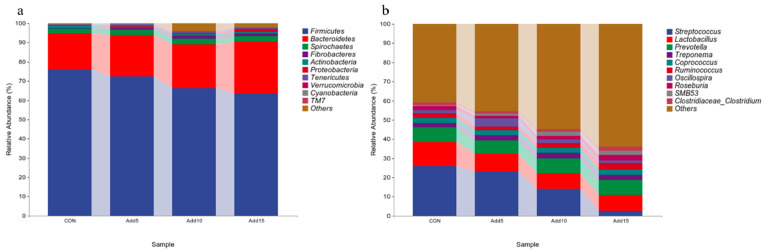
The relative abundance of bacteria in each group. (**a**) Phylum level. (**b**) Genus- level.

**Figure 4 animals-14-00662-f004:**
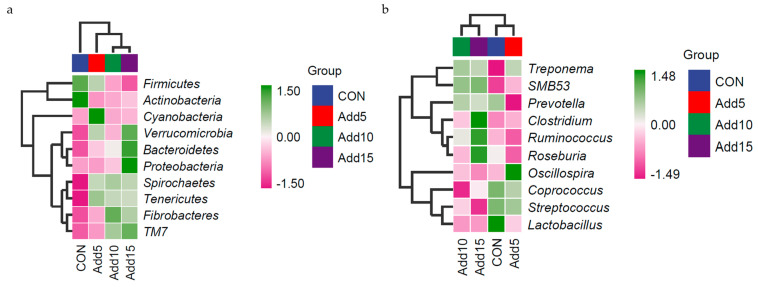
Heatmaps showing the taxonomic composition of Hezuo pigs at four wholeplant corn silage addition ratios at the phylum (**a**) and genus (**b**) levels. Colors in the heatmap indicate normalized (log_10_-transformed) relative abundance; red to blue represent most abundant to least abundant.

**Table 1 animals-14-00662-t001:** The levels of the conventional nutrient contents in whole-plant corn silage (DM basis).

Items	Content (%)
Crude protein	11.36
Ether extract	4.66
Neutral detergent fiber	41.29
Acid detergent fiber	29.29
Crude ash	35.49
Calcium	2.47
Phosphorus	0.32

**Table 2 animals-14-00662-t002:** Composition and nutrient levels of the experimental diets (DM basis).

Items	CON	Whole-Plant Corn Silage Addition Level (%)		
		5	10	15
Ingredients (%)				
Corn	69.00	66.10	63.20	60.30
Soybean meal	20.00	19.90	19.80	19.70
4% Premix ^(1)^	4.00	4.00	4.00	4.00
Wheat bran	7.00	5.00	3.00	1.00
Whole-plant corn silage	0.00	5.00	10.00	15.00
Total	100.00	100.00	100.00	100.00
Nutrient levels ^(2)^				
Metabolizable energy (MJ/kg)	13.47	13.35	13.23	13.10
CP (%)	15.89	15.74	15.60	15.46
CF (%)	3.45	4.65	5.85	7.05
Ca (%)	0.50	0.62	0.74	0.85
P (%)	0.35	0.35	0.35	0.36
Available phosphorus (%)	0.10	0.10	0.11	0.11
NDF (%)	11.80	12.83	13.87	14.91
ADF (%)	5.25	6.34	7.43	8.53
Lysine (%)	0.85	0.83	0.82	0.80
Threonine (%)	0.58	0.57	0.56	0.55
Methionine (%)	0.24	0.23	0.23	0.23

^(1)^ Premix provided the following per kg of diet: Fe (as ferrous sulfate) 2~7 g, Cu (as cupric oxide) 0.200–0.625 g, Zn (as zinc sulfate) 1.0–2.0 g, Mn (as manganese sulfate) 0.5–1.5 g, Se (as sodium sulfate) 3.0–10.0 mg, I 3.0–25.0 mg, VA 100–160 KIU, VD 30–80 KIU, VE 290 mg, VK3 0.0–90.0 mg, VB_,_ 25 mg, VB_6_ 25 mg, VB_12_ 0.2 mg, pantothenic acid 180 mg, nicotinic acid 240 mg, and folic acid 10 mg; ^(2)^ nutrient levels are calculated values. CP: crude protein; CF: crude fiber; NDF: neutral detergent fiber; ADF: acid detergent fiber.

**Table 3 animals-14-00662-t003:** The effects of whole-plant corn silage on the growth performance of Hezuo pigs.

Items	CON	Whole-Plant Corn Silage Addition Level (%)		
		5	10	15
IBW (kg)	8.06 ± 0.25	7.78 ± 0.37	7.56 ± 0.40	8.12 ± 0.18
FBW (kg)	39.60 ± 1.17	40.81 ± 2.74	43.36 ± 1.93	39.62 ± 2.07
ADFI (kg/day)	1.08 ± 0.04	1.10 ± 0.06	1.16 ± 0.05	1.07 ± 0.05
ADG (kg/day)	0.26 ± 0.01	0.28 ± 0.02	0.30 ± 0.02	0.26 ± 0.02
F/G ratio	4.11 ± 0.09 ^a^	4.01 ± 0.07 ^ab^	3.89 ± 0.03 ^b^	4.10 ± 0.08 ^ab^

In the same row, values with different superscript letters differ significantly (*p <* 0.05). IBW: initial body weight; FBW: final body weight; ADFI: average daily feed intake; ADG: average daily gain; F/G ratio: feed-to-gain ratio.

**Table 4 animals-14-00662-t004:** The effects of whole-plant corn silage on serum biochemical indices of Hezuo pigs.

Items	CON	Whole-Plant Corn Silage Addition Level (%)		
		5	10	15
TP (g/L)	59.61 ± 2.11 ^b^	64.77 ± 1.47 ^ab^	66.82 ± 1.66 ^a^	63.46 ± 1.49 ^ab^
ALB (g/L)	27.93 ± 0.62 ^b^	30.16 ± 0.37 ^ab^	32.82 ± 0.18 ^a^	30.54 ± 2.13 ^ab^
TC (mmol/L)	3.01 ± 0.10	2.97 ± 0.08	2.92 ± 0.36	2.85 ± 0.36
TG (mmol/L)	0.77 ± 0.01 ^b^	0.81 ± 0.05 ^ab^	0.87 ± 0.01 ^a^	0.80 ± 0.03 ^ab^
HDL-C (mmol/L)	0.76 ± 0.02	0.78 ± 0.01	0.80 ± 0.05	0.76 ± 0.02
LDL-C (mmol/L)	1.19 ± 0.26	1.11 ± 0.24	1.18 ± 0.31	1.39 ± 0.16
Cr (μmol/L)	84.35 ± 4.59	80.47 ± 4.24	77.92 ± 3.02	80.60 ± 4.18
Urea (μmol/L)	6.26 ± 0.25	5.84 ± 0.17	6.22 ± 0.26	6.58 ± 0.19
GLU (mmol/L)	3.85 ± 0.18 ^b^	4.55 ± 0.02 ^ab^	4.80 ± 0.17 ^a^	4.11 ± 0.44 ^ab^
AST (U/L)	62.49 ± 2.35	56.87 ± 0.94	55.41 ± 2.32	59.02 ± 2.66
ALT (U/L)	54.02 ± 0.58	56.90 ± 3.52	59.21 ± 0.90	56.05 ± 1.25

In the same row, values with different superscript letters differ significantly (*p <* 0.05). TP: total protein; ALB: albumin; TC: total cholesterol; TG: triglyceride; HDL-C: high-density lipoprotein cholesterol; LDL-C: low-density lipoprotein cholesterol; Cr: creatine; GLU: glucose; AST: aspartate aminotransferase; ALT: alanine aminotransferase.

**Table 5 animals-14-00662-t005:** The effect of whole-plant corn silage on the alpha diversity of the fecal microorganisms of Hezuo pigs.

Items	CON	Whole-Plant Corn Silage Addition Level (%)		
		5	10	15
Chao1 index	1635.86 ± 129.71	1715.04 ± 126.08	1984.53 ± 80.11	1950.31 ± 104.02
Observed index	1544.82 ± 133.14	1620.38 ± 118.79	1891.58 ± 80.88	1890 ± 106.08
Shannon index	7.36 ± 0.23 ^c^	7.87 ± 0.15 ^bc^	8.24 ± 0.14 ^ab^	8.79 ± 0.22 ^a^
Simpson index	0.94 ± 0.01 ^b^	0.97 ± 0.01 ^a^	0.97 ± 0.01 ^a^	0.99 ± 0.01 ^a^

In the same row, values with different superscript letters differ significantly (*p <* 0.05).

**Table 6 animals-14-00662-t006:** The taxonomic composition of the top 10 most abundant fecal microbial communities at the phylum level.

Phylum	CON	Whole-Plant Corn Silage Addition Level (%)		
		5	10	15
Firmicutes	76.13 ± 1.36 ^a^	72.36 ± 1.68 ^ab^	66.28 ± 3.07 ^ab^	63.40 ± 5.29 ^b^
Bacteroidetes	18.65 ± 1.12 ^b^	21.47 ± 1.82 ^ab^	22.75 ± 1.93 ^ab^	27.02 ± 3.84 ^a^
Spirochaetes	2.17 ± 0.89	2.87 ± 0.33	2.94 ± 0.98	2.89 ± 0.83
Fibrobacteres	0.45 ± 0.23	0.77 ± 0.11	1.59 ± 0.65	1.31 ± 0.48
Actinobacteria	1.19 ± 0.32	0.66 ± 0.06	0.69 ± 0.10	0.74 ± 0.16
Proteobacteria	0.48 ± 0.13 ^b^	0.47 ± 0.05 ^b^	0.64 ± 0.18 ^b^	1.69 ± 0.49 ^a^
Tenericutes	0.36 ± 0.07 ^b^	0.79 ± 0.07 ^a^	0.73 ± 0.14 ^a^	0.73 ± 0.13 ^a^
Verrucomicrobia	0.07 ± 0.01 ^c^	0.15 ± 0.02 ^ab^	0.10 ± 0.03 ^bc^	0.18 ± 0.03 ^a^
Cyanobacteria	0.06 ± 0.02 ^b^	0.13 ± 0.02 ^a^	0.06 ± 0.01 ^b^	0.06 ± 0.02 ^b^
TM7	0.02 ± 0.01	0.04 ± 0.01	0.11 ± 0.07	0.13 ± 0.07
Others	0.43 ± 0.05	0.29 ± 0.05	0.49 ± 0.11	0.53 ± 0.10

In the same row, values with different superscript letters differ significantly (*p <* 0.05).

**Table 7 animals-14-00662-t007:** The taxonomic composition of the top 10 most abundant gut microbial communities at the genus level.

Genus	CON	Whole-Plant Corn Silage Addition Level (%)		
		5	10	15
*Streptococcus*	25.93 ± 4.32 ^a^	23.30 ± 2.24 ^a^	14.04 ± 3.26 ^b^	2.69 ± 2.03 ^c^
*Lactobacillus*	12.54 ± 3.64	8.96 ± 1.34	8.17 ± 1.79	8.19 ± 2.57
*Prevotella*	7.81 ± 1.96	6.95 ± 0.95	7.76 ± 0.94	7.68 ± 1.19
*Treponema*	2.15 ± 0.89	2.85 ± 0.33	2.91 ± 0.97	2.85 ± 0.81
*Coprococcu* *s*	2.70 ± 0.47	2.68 ± 0.27	2.55 ± 0.35	2.64 ± 0.50
*Ruminococcus*	2.23 ± 0.30 ^b^	1.84 ± 0.14 ^b^	2.61 ± 0.45 ^ab^	3.37 ± 0.35 ^a^
*Oscillospira*	1.76 ± 0.37 ^b^	4.14 ± 0.70 ^a^	1.84 ± 0.45 ^b^	1.35 ± 0.22 ^b^
*Roseburia*	2.08 ± 0.67 ^ab^	1.34 ± 0.47 ^b^	1.77 ± 0.35 ^ab^	2.97 ± 0.44 ^a^
*SMB53*	0.65 ± 0.13 ^b^	1.23 ± 0.19 ^ab^	2.07 ± 0.40 ^a^	2.17 ± 0.49 ^a^
*Clostridiaceae_Clostridium*	1.16 ± 0.16	1.28 ± 0.13	1.34 ± 0.10	2.04 ± 0.74
Others	40.99 ± 2.01 ^c^	45.43 ± 1.51 ^bc^	54.93 ± 3.90 ^ab^	64.04 ± 3.98 ^a^

In the same row, values with different superscript letters differ significantly (*p <* 0.05).

## Data Availability

The sequence files determined in the present study were deposited at the Sequence Read Archive (SRA; http://www.ncbi.nlm.nih.gov/subs/ (accessed on 27 December 2023); SRA accession number: PRJNA1056331).

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
