# Peer review of "Effects of Whole-Plant Corn Silage on Growth Performance, Serum Biochemical Indices, and Fecal Microorganisms in Hezuo Pigs"

_animals, 2024, doi:10.3390/ani14050662_

Round 1

Reviewer 1 Report

Comments and Suggestions for Authors

The selected topic holds scientific significance, with this investigation concentrating on the systematic evaluation of the repercussions ensuing from the inclusion of varying proportions (5%, 10%, and 15%) of whole-plant corn silage within the dietary regimens of Hezuo pigs. The study delineates the impact associated with the integration of whole-plant corn silage into the dietary matrix of pigs, particularly elucidating its potential to positively modulate growth, metabolism, and gastrointestinal health. These insights yield practical implications for the refinement of porcine nutritional strategies within the broader ambit of livestock management. While the discussion adeptly traverses the principal findings and their interpretation, there remains for refinement. In conclusion, following requisite modifications, the study could be published.

Specific line comments:

L36: Add “the” before “principal”

Lines 47-89: I suggest you add references in the text. As in line 59 and 82.

Line 70: are there recent study in other species?

Line 86: Why have you chosen these percentage?

Line 92: Could be possible that the weather can change something in your study?

Lines 303-433: I personally found the discussion very long. In addition, I suggest adding a few references to enforce the value of your results.

Lines 435-441: The conclusion are too ambitious, are really “notable positive” only the inclusion 10% ? The growth performance not strongly changing.

Line 473 : It’s absent the year of the research.

Line 475: You have to follow the journal reference’s instructions, is it year position in the sentence, correct?

Line 524: can you modify the year characters?

Reviewer 2 Report

Comments and Suggestions for Authors

Dear Author

You have recently submitted the manuscript entitled “Effects of whole-plant corn silage on growth performance, serum biochemical indices, and fecal microorganisms in Hezuo pigs” The manuscript is written in poor quality, and there are a lot of grammatical and technical mistakes.

The simple summary is poorly written and did not address the issue of the study. In the introduction section many statements are blind. The most crucial thing is, what is the correlation between the indicators measured throughout the text? Why is 10% the appropriate level? It seems that this conclusion cannot be drawn from the results and discussion expressions of this article. It is difficult to clearly express the relationship between fecal microbiota, growth performance, blood indicators, and experimental treatment from the manuscript. So, I don't think the current status of the manuscript can be accepted.

The manuscript still has other issues, such as:

Line 47-51: the sentence “The livestock sector is the backbone of the global agricultural economy and China is currently the largest livestock producer worldwide. Pig production accounts for a large proportion of this economy, with China, the European Union, the United States, Brazil, and Canada being the leading regions pig stocks and pork production”. These sentences seem to have little relevance to the purpose of this study

Line 60-62: “China is currently facing a serious food shortage problem. In 2021, the total grain supply in China was 870 million tons, 450 million of which were consumed by the farming industry

,accounting for 51.72% of the total consumption.”, without references.

Line 73-76: There are significant differences in digestive physiology between monogastric animals and ruminant animals, and these sentences seem to have little relevance to the research content of this manuscript.

Line 78-79: “The effects of whole-plant corn silage have mainly been assessed in ruminants such

as dairy cows and beef cattle, and little is known about how they affect monogastric animals.” What aspects affect monogastric animals?

Line 97: Are the levels of nutrient contents in whole-plant corn silage in Table 1the actual measured values?Before or after sun drying?This is crucial for experimental design.

Line 149: Please check if the DNA Kit is correct.

Line 164-172: This section “The 16S rRNA gene sequencing data……....Heatmap A heatmap was plotted using R package 3.5.1” should be part of the data analysis.

Line 174-181: what do you mean by “(Shannon index, which measures uncertainty …….community diversity [13])”. The explanation appears redundant, and the references seems inappropriate.

Line 116: Table 3 ADFI (kg) should be ADFI (kg/day)? ADG (kg) be ADG (kg/day)?

Table 4: “Item” or “Items”? Please check all tables.

Line 210: what do you mean by “gut microflora”? fecal microflora?

Line 231-237: what do you mean by beta diversity between Unweighted UniFrac and weighted UniFrac in the sentence ? What do you want to express about the result?

Comments on the Quality of English Language

The manuscript is written in poor quality, and there are a lot of grammatical and technical mistakes.

Round 2

Reviewer 2 Report

Comments and Suggestions for Authors

I do not think that the simple summary can be "abstract of the abstract", so it was still poorly written. The simple summary must explain the expected impact that the results may have on practice, when they will be applied. Impact may be economic, environmental or social. The simple summary should not be limited to presenting the context and objectives. They are written in simple English suitable for non-specialists or even non-science readers.

 I still believe that the manuscript did not clearly state the relationship and interrelationships between fecal microbiota, growth performance, blood indicators, and experimental treatment.

Comments on the Quality of English Language

Minor editing of English language required

Author Response

Dear Reviewers:

We would like to thank you very much for your valuable comments and good suggestions that greatly helped to improve our manuscript. We have carefully considered your valuable comments and good suggestions. In the following, we are going to explain how your comments have been taken into full account in the revision.

  1. I do not think that the simple summary can be "abstract of the abstract", so it was still poorly written. The simple summary must explain the expected impact that the results may have on practice, when they will be applied. Impact may be economic, environmental or social. The simple summary should not be limited to presenting the context and objectives. They are written in simple English suitable for non-specialists or even non-science readers.

Answer: Thank you for your valuable comment. We have addressed the issue in the revised manuscript. Responses are following:

In recent years, with the expansion of pig husbandry, the competition for grain between its use as feedstuffs for animals and as foodstuffs for humans has become increasingly prominent. Tradition-al feeds such as corn, soybean, and wheat bran are no longer sufficient to meet the growing needs of the pig industry. Consequently, it is imperative to develop novel, unconventional feedstuffs. Whole-plant corn silage has the advantages of high energy, easy digestion, good palatability, aro-matic taste, long preservation time, and low cost. In this study, we found that adding proper amount of whole-plant corn silage in the diet can not only promote the growth of Hezuo pigs, but also replace part of the traditional feed. Whole-plant corn silage has a broad application prospect in Hezuo pig diet.

  1. I still believe that the manuscript did not clearly state the relationship and interrelationships between fecal microbiota, growth performance, blood indicators, and experimental treatment.

Answer: Thank the reviewer for the valuable suggestion. This study was conducted to investigate the effects of different proportions of whole-plant corn silage in diets on growth performance, serum biochemical indices and fecal microorganisms of Hezuo pigs. The results showed that adding 10% whole-plant corn silage had the highest FBW and the lowest feed/meat ratio. In addition, the contents of serum biochemical indexes such as TP, ALB, TG, and CLU were dramatically elevated. There was a positive correlation between fecal microbial diversity and the proportion of whole-plant corn silage added. Notably, the increase in fecal microbial diversity did not affect the growth of Hezuo pigs. In addition, we found that with the increase of whole-plant corn silage supplemental level, the number of streptococcus decreased significantly, which indicates that the addition of whole-plant corn silage in feed can reduce the abundance of certain pathogenic bacteria in pigs, thereby reducing the incidence of disease in animals and improving the efficiency of pig breeding. The above research results show that the scheme of adding whole-plant corn silage is feasible. We added and annotated in green in the discussion section.